# Crystallization Behavior and Physical Properties of Monoglycerides-Based Oleogels as Function of Oleogelator Concentration

**DOI:** 10.3390/foods12020345

**Published:** 2023-01-11

**Authors:** Yingzhu Zhang, Jinqi Xu, Cuie Tang, Yan Li

**Affiliations:** 1College of Food Science and Technology, Huazhong Agricultural University, Wuhan 430070, China; 2Key Laboratory of Environment Correlative Dietology, Huazhong Agricultural University, Ministry of Education, Wuhan 430070, China; 3Functional Food Engineering & Technology Research Center of Hubei Province, Wuhan 430070, China

**Keywords:** oleogel, monoglyceride, chain length, crystallization, physical properties

## Abstract

Oleogels have been shown as a promising replacer of hydrogenated vegetable oil. Fatty acid glycerides, including some typical mono- and di-glycerides, were used to form oleogels. The concentration effects of fatty acid glycerides on the crystallization behavior and physical properties of oleogels were investigated by using different analysis techniques. The results showed that all the oleogels formed by saturated fatty acid glycerides (glyceryl monostearate (GMS), glyceryl monolaurate (GML), glycerol monocaprylate (GMC)) exhibited a solid-like behavior and were thermally reversible systems, while a higher amount of unsaturated fatty acid glycerides (monoolein (GMO), diolein (GDO)) were needed to form oleogels. The onset gelation concentration of GMS and GMC was found to be 2 wt% (*w*/*w*), while that of GML was 4 wt% by the inverted tube method. The crystallization results illustrated that the GMS and GMC formed small needle-like crystals with the presence of β and β′ crystals, while GML formed large flake-like crystals with α crystals in oleogels, and faster cooling rates caused smaller crystals. GMS- and GMC-based oleogels had higher crystallinity, resulting in higher thermal stability and better mechanical properties than GML-based ones at the same monoglyceride (MAG) level. With the increasing MAG content, the oleogels showed a more compact three-dimensional network leading to higher mechanical properties and better thermal stability and resistance to deformations. Hence, MAG-based oleogels, especially GMC ones with medium chain fatty acid, could be a promising replacer for hydrogenation vegetable oils.

## 1. Introduction

Solid or semi-solid fats are widely used in the food industry due to their excellent texture, stability, and plasticity properties. To increase production and reduce costs, hydrogenation of vegetable oils has been used for decades to transform liquid oils into solid or semi-solid fats, which may increase the content of saturated and trans fatty acids [1]. It has been reported that the excessive intake of saturated and trans fats in daily diet can cause adverse effects on health [2]. Hence, numerous approaches have been developed to convert vegetable oils into solid or semi-solid fats with low saturated and trans fatty acids content. Oleogels have been indicated as a viable alternative with compatible properties for food applications [1,3]. Oleogel is a thermally reversible viscoelastic liquid or solid lipid mixture composed of liquid oil and oleo-gelator, in which the oleo-gelator forms a three-dimensional network structure through self-assembly or crystallization to inhibit the flow of liquid oil [4,5]. According to the latest view of Guenet, a gel should meet two more criteria involving the topology and the thermodynamic properties [6]. Up to now, many different oleo-gelators have been developed, such as monoglycerides [7], diglycerides [8], hydroxypropyl methyl cellulose [9], ethyl cellulose [10], biological wax [11,12], phospholipids [13], stearic acid [14], phytosterols [14]. Additionally, the application of oleogel systems in replacing solid fat [9,15], transporting active substances [16] and controlling in vitro digestion [11] has also been well demonstrated.

Among the existing oleo-gelators, oleogels formed by monoglycerides (MAGs) are some of the most promising options because MAGs can crystallize and form strong networks with good elastic properties even at low concentrations (3%) [17]. In the hydrophobic medium, the glycerol head of MAG self-assembles to form an inverse lamellar phase, and hydrogen bonds established between hydrophilic head groups lead to the crystallization of the MAG aliphatic tails, resulting in oil gelation and providing elasticity to the MAG-oil system [18,19,20,21]. Therefore, the rheological, textural, and thermal properties of oleogels are largely affected by the MAG crystallization behavior, which in turn is influenced by the oil types and processing conditions such as concentration, temperature, and cooling rate [22,23,24]. Generally, a dense or loose crystal structure can be constructed by adjusting the concentration of the gelators, so as to obtain oleogels with the desired properties. As reported, the GML molecules crystallized and formed a tighter three-dimensional network with increasing concentration, leading to higher thermal stability and higher deformation resistance of the oleogels [25]. The modulation of the crystallization behavior of GMS and commercial monoglycerides (59.78% stearic acid) on the properties of oleogels at different concentrations has also been well demonstrated [17,26]. More importantly, the composition of MAG molecules and the length of the hydrocarbon chain affect the self-assembly behavior of MAG, causing different crystallization behavior of oleogels [27]. Oleogels composed of commercial GMS and monopalmitin (GMP) exhibited smaller crystals and higher rheological properties than pure GMS-based oleogels [18]. The oleogels prepared by glycerol monobehenate (GMB) with longer chain length had smaller needle-like and platelet-like crystals, leading to the highest properties and stability than GML and GMS-based oleogels [28]. According to the existing findings, it was proposed to modulate the crystallization behavior of oleogels by combining the concentration of fatty acid glycerides and environmental conditions like temperature. We also aimed to find out the potential use of glyceride with medium chain of fatty acid.

Hence, we selected several fatty acid glycerides with different carbon chain lengths and saturation degrees. First, the onset gelation concentration to form oleogel was tested. Then, the physical properties of oleogels were determined. After that, the crystallization behavior and thermal properties of oleogels were investigated by means of polarized light microscopic (PLM), X-ray diffraction (XRD), and differential scanning calorimetry (DSC) analysis. The results of this study may provide a reference to tailor the crystallization behavior and physical properties of oleogels by adjusting the composition and concentration of MAG and temperature conditions.

## 2. Materials and Methods

### 2.1. Materials

Glyceryl monostearate (GMS, analytical pure), glycerol monolaurate (GML, ≥90% purity), and glycerol monocaprylate (GMC, analytical pure) were all purchased from Yuanye Biotechnology Co., Ltd. (Shanghai, China). Diolein (GDO, 90% purity) and monoolein (GMO, ≥50% purity) were purchased from Aladdin Biochemical Technology Co., Ltd.(Shanghai, China). Soybean oil was purchased from a local market in Wuhan, China.

### 2.2. Preparation of Oleogels

Different amounts of fatty acid glycerides were added into soybean oil and heated to 80 °C with agitation at 500 rpm for complete dissolution. Then, the mixtures were cooled down to room temperature and equilibrated at 4 °C for 24 h until analyses.

### 2.3. Determination of Onset Gelling Concentrations (OGC) 

OGC was defined as the lowest concentration of fatty acid glycerides required to form solid-like oleogels, which was tested by the inverted tube method. According to the method in Section 2.2, fatty acid glycerides with different amounts (0.5–15 wt%) was added into the oil phase. After being placed at 4 °C for 24 h, the sample bottles were taken out and inverted at room temperature for 1 h to test the oleogel formation.

### 2.4. Characterization of Oleogels

#### 2.4.1. Polarized Light Microscopy (PLM)

The microstructure of oleogel samples was visually observed under a polarized light microscope (MG-100, M-shot, Guangzhou, China) equipped with a digital camera. After 24 h of storage, the oleogel sample was lightly spread on the microscope glass slide, and a cover slip was placed above immediately to ensure uniformity. The microstructure was observed at 25 °C using a 20× objective lens. 

#### 2.4.2. Hardness and Oil Binding Capacity (OBC)

The OBC of oleogel samples was evaluated using a centrifugation method proposed by Giacomozzi et al. [17]. Briefly, about 1.0 g of oleogel sample was placed in a centrifuge tube and then centrifuged at 25 °C using a microcentrifuge (Microfuge 20R, BECKMAN COULTER, Pasadena, CA, USA). After that, the centrifuge tube was inverted for 5 min, and the released oil was removed with filter paper. The OBC of oleogels was determined according to Equation (1):(1)$$\mathrm{OBC}\left(\%\right)=100-\left(\frac{\mathrm{weight}\;\mathrm{of}\;\mathrm{released}\;\mathrm{oil}}{\mathrm{weight}\;\mathrm{of}\;\mathrm{oleogel}}\right)\times 100$$

The hardness of oleogel samples was measured by TA. XT Plus texture analyzer (Stable Micro Systems, London, UK). In detail, a P/6 cylindrical probe was selected, and the parameters were set as follows: the pre-test speed was 1 mm/s, the test speed was 2 mm/s, the post-test speed was 2 mm/s, the trigger force was 5 g and the puncture distance was 10 mm. The maximum force during the TA test was considered as hardness.

#### 2.4.3. Rheology Measurements

A rheometer (DHR2, TA, New Castle, DE, USA) equipped with a Peltier plate temperature controller was used to analyze the rheological properties of oleogel samples. A 40 mm aluminum plate configuration was selected, and the gap was maintained at 1000 μm. The stain sweep test was first carried out at a frequency of 1 Hz to investigate the linear viscoelastic region (LVR) of oleogels, and then, the frequency sweep test was performed in a range of 0.1–100 Hz with a strain of 0.1%. After that, the change in the apparent viscosity of oleogels with shear rates (0.01–100 s^−1^) was evaluated. All the above tests were conducted under 25 °C. Finally, temperature sweep tests, including two sweeps (heating-cooling) between 5 °C and 70 °C at a rate of 2 °C/min were carried out under 1 Hz and 0.1% strain to investigate the effects of temperature on the rheological properties of the oleogel samples.

#### 2.4.4. Differential Scanning Calorimetry (DSC)

The melting behavior of oleogel samples was determined by differential scanning calorimetry system (DSC 204 F1, NETZSCH, Selb, Germany). Approximately 8–10 mg oleogel sample was weighed and encapsulated in an aluminum crucible. An empty sealed aluminum crucible was used as a reference. The oleogel samples were first analyzed at a heating rate of 30 °C/min from 25 °C to 80 °C and held for 10 min to eliminate crystal memories. After that, the samples were cooled down to 0 °C and then heated to 80 °C at a rate of 10 °C/min.

#### 2.4.5. X-ray Diffraction (XRD)

The crystal polymorphism of oleogel samples were studied using X-ray diffractometer (D8 Advance, Bruker, Karlsruhe, Germany) with a Cu-Kα source (λ= 1.54 Å) operated at 40 kV and 30 mA. The samples were analyzed at a diffraction angle range from 3° to 50° (2θ) at 25 °C with a step size of 0.02°. Data processing and analyses were performed using MDI Jade software (Version 6.0, Materials Data Inc., Livermore, CA, USA). 

#### 2.4.6. Crystal Morphology during Heating and Cooling Process

Morphology changes of crystals during the heating and cooling process was observed by a polarized light microscope (MG-100, M-shot, Guangzhou, China) equipped with a temperature controller (mK2000, INSTEC, Boulder, CO, USA). A small amount of oleogel samples was placed on the stage of the thermostat and heated from 20 °C to 80 °C, maintained at 80 °C for 10 min, and then cooled to 20 °C with the rates of 2 °C/min, 10 °C/min and 20 °C/min to simulate slow, medium, and fast cooling, the crystal morphology at different rates and temperatures were recorded. 

## 3. Results and Discussion

### 3.1. Oleogel Formation and Microstructure Observation

In a thermos-reversible gel, one of the miscible components will undergo a liquid-solid transition upon the onset gelation concentration during the cooling process [6]. Hence, the gelling ability of several fatty acid glycerides was tested, and it was found that liquid oil could not be fixed due to the weak crystallization behavior of unsaturated glycerides (GMO and GDO) even at high concentrations (15%), even though the crystal structure could be observed under the polarized light microscopy (Appendix A). Saturated fatty acids glycerides could form solid-like oleogels well at relatively low concentrations (Figure 1). The results showed that at least 2 wt% of GMS and GMC and 4 wt% GML were needed to produce a semi-solid structure that did not flow when the test tubes were inverted. Therefore, it can be considered that the OGC of oleogel formed by GMS and GMC with soybean oil is around 2 wt%, and that of GML is 4 wt%. To further study the effect of oleo-gelator on the physical properties of oleogels, the gel samples containing 4 wt%, 6 wt%, 8 wt%, 10 wt% of saturated MAGs were used as the object of subsequent characterization. 

The crystal structure of oleogels depends on the concentration and composition of fatty acid glycerides (Figure 1 and Appendix A). The morphology of GMO crystals in the mixture system was like clusters, which is more like a spherulitic structure. While the crystals in GMS and GMC oleogels are needle-like and uniform dispersion, and those in GML oleogels presented large flake-like crystals, which were much larger than those in GMS and GMC oleogels. Similar needle-like GMS and large fiber-like GML oleogel crystal morphologies have been reported in previous studies [22,23]. The difference in crystallization behavior of MAGs is related to lipid crystallization kinetics and nucleation mechanism, which is affected by chain length, saturation, and molecular symmetry of glycerides [29]. Li et al. reported that the chain length is related to the dimension of crystal growth and nucleation mechanism, and they found that for pure MAG oleogels, the crystal growth mode changed from spherulitic growth to rod-like growth with chain length increasing [28]. At low MAG concentration, GML crystals did not intertwine together, and the sample behaved like a paste, while GMS and GMC crystals linked to form a loose network structure. With the increasing concentration of MAGs, the number of crystals increased, the cross-linked crystals became denser, and a gelling network formed by crystals was observed. This is consistent with the results of Pan et al. [22], who demonstrated that higher GML concentration in the gel lead to a more compact and denser network structure. Therefore, in this study, the difference in crystal morphology of oleogels may be caused by the difference in the length and structure of the carbon chain of fatty acid glycerides. It is worth noting that GMC with the shortest chain length of fatty acid could also form crystals well, similar to GMS.

### 3.2. Oil Binding Capacity and Mechanical Strength

The crystallization behavior would affect the physical properties of oleogels. Table 1 presents that the OBC of oleogels increased as the gelator concentration increased. When the proportion of MAG increased from 4% to 10%, OBC of oleogels greatly increased from 79.88% to 98.06%, 77.06% to 93.75% and 46.51% to 74.85% for GMS, GMC, and GML, respectively, which may be because more gelators were involved in the construction of crystalline networks, thus reducing the oil loss, and increasing the mechanical strength of the system. In addition, the OBC of GMS and GMC were found to be higher than GML at the same concentration. Generally, the number, size, morphology, and spatial distribution of crystals in the network are critical factors to affect the physical properties of gels [30]. The needle-like and fibrous crystals with a high surface area can form interaction forces with more liquid components, thus helping to improve the physical stability of the oleogel [31]. As aforementioned, the crystal structure formed by smaller acicular crystals in GMS and GMC oleogels would have a larger surface area and a more zigzag pathway than the larger flake-like crystals in GML oleogels, thereby preventing oil migration and consequent leakage [28,32]. Lopez-Martinez et al. also reported that the three-dimensional crystal organization developed by smaller sub-α crystals would provide a larger surface area than those formed from bigger crystals, thus providing a more efficient structure for oil entrapment [18].

Meanwhile, the hardness of oleogels was determined using a texture analyzer. The increase of MAG concentrations, the hardness increased significantly (Table 1). In addition, the hardness values are also affected by MAG types. The GMS oleogels led to the highest hardness, while GML oleogels possessed the smallest hardness values. Judging from the previous discussion, it is speculated that the looser crystal networks caused by its larger fibrous crystals are the reason for the lower hardness of GML oleogels. These results showed that GMS and GMC oleogels had better mechanical properties.

### 3.3. Rheological Analysis

The linear viscoelastic region (LVR) of oleogel samples was first determined (Appendix A). Subsequently, the frequency sweep and steady shear test were conducted within the determined LVR strain values 0.1%.

As shown in Figure 2a, for all oleogel samples, the storage moduli (G′) were higher than the loss moduli (G″) without overlapping in the frequency range, indicating that all oleogels were viscoelastic materials with solid-like behavior [33]. For GMS and GMC oleogels, the slope of G′ curve approached 0 and G″ became minimum at an intermediate frequency. It was reported that when G′ is plotted as a function of frequency, the oleogel samples observed exhibit a frequency-independent behavior, which shows a connection state of the crystals inside the system, indicating a strong gel behavior [34]. As for GML oleogels, both G′ and G″ almost paralleled each other in the frequency range of 0.1–100 Hz, indicating a weak gel state [34,35]. Meanwhile, G′ and G″ of oleogels increased with the increase of gelator addition, demonstrated that higher concentration of gelator accelerated the formation of oleogel network structure and improved the rheological properties of the system. We also found that the G′ and G″ of GMS and GMC oleogel samples were higher than those of GML oleogels under the same concentration and measurement conditions, indicating the stronger gel network of GMS and GMC oleogels than GML oleogels. Figure 1 also showed that the interaction of GML crystals was weaker than that of GMS and GMC crystals, which caused the loose gel network.

Figure 2b exhibits that the viscosity of all the oleogel samples decreased with an increase in the shear rate, indicating the shear-thinning behavior [36] and the breakdown of the gel network during shearing. Most non-Newtonian food systems possess shear-thinning behavior [37], as observed in many oleogel systems [38]. The viscosity also showed a tendency to increase with increasing levels of three MAGs, likely due to more MAG crystallized to entrap the oil molecules resulting in a stronger ability to resist shear rates. In addition, the viscosity of GMS- and GMC-based oleogels was higher than that of GML samples, which were in good accordance with the results in Figure 2a.

### 3.4. Thermal Analysis

Temperature sweep tests were conducted to evaluate the stability in response to heat treatment of the samples. As shown in Figure 3a, G′ and G″ of oleogels remained relatively stable at the initial stage of melting. As the temperature continued to increase, both the G′ and G″ values of oleogels decreased rapidly, indicating the breakdown of the crystalline network structures. Subsequently, the gelled state was lost at the crossover point (G′ = G″), and the oleogels completely transformed into liquid, exhibiting a viscous behavior at high temperatures. The crossover point was reached at around 50–60 °C for GMS and GMC oleogels and 35–45 °C for the GML samples and gradually moved to high temperature as their concentration increased, which indicated that the oleogels formed by GMS and GMC have higher thermal stability. The observation of higher thermal stability of oleogels with higher gelator concentration is consistent with the results reported by [39].

As the temperature decreased, the oleogel systems approached phase transition until reaching the temperature of crystallization. A dramatic increase of G′ and G″ was found in the oleogel system (Figure 3b), which is because the interaction between the crystalline aggregates during the cooling process extends to the entire oleogel system, forming a three-dimensional network structure again [24]. The temperature sweep results showed that the MAG oleogel is a thermally reversible system. However, for GML oleogels, due to their low gel strength, the structural recovery ability during cooling was relatively weak.

The thermodynamic properties of the oleogel system were further analyzed by DSC (Appendix A and Appendix A). The melting curves for MAG oleogels with a concentration of 6% are presented in Figure 4a. The samples with GMS and GMC exhibited two endothermic peaks in the melting curves, in agreement with [24]. The primary endothermic peaks of 6% GMS and 6% GMC oleogel samples were located at 49.10 °C and 50.05 °C, respectively, while the lower primary peak value was measured at 6% GML, locating at 44.35 °C. It was reported that the melting and crystallization peak temperature of MGs was dependent on acyl chain length [18]. That means the value of these peaks decreases by decreasing the chain length, and hence the melting peak of GML is lower than that of GMS. Interestingly, GMC had a similar peak value of GMS. In MAG oleogel system, the dominant peak was connected with the melting of Lα phase, and the secondary peak was related to the transformation of Lα and sub-α [28]. However, GML oleogels displayed only one single endothermic peak at around 40 °C, which might indicate that there is only melting of Lα phase during heating treatment. Therefore, it is speculated from the melting behavior that the polymorphism of crystals in GMS and GMC oleogels is different from that in GML oleogels. The enthalpy (ΔH) evaluated by the area under the peak is an estimate of the amount of crystallized material in the sample [40], and a higher enthalpy value indicates a higher degree of crystallinity [23]. ΔH of 6% GMS and 6% GMC were nearly 5.26 and 5.18 W/g, respectively, while the 6% GML had a smaller ΔH value of 3.85 W/g. This can be attributed to the weaker network formed by GML inside the oleogels, which was supported by the microstructure, OBC, and hardness results. Overall, GMS and GMC oleogels possess higher crystallinity, which causes higher heat resistance and better mechanical properties than GML-based oleogels. The primary peaks gradually moved to a higher temperature, and the intensity of this peak increased with an increase in gelator concentration (Appendix A), consistent with the results in Figure 3. Furthermore, the melting enthalpy of oleogels increased with increasing MAG content, indicating the presence of enhanced structural systems [17].

### 3.5. XRD Analysis

The polymorphism of oleogels was evaluated by X-ray diffraction (XRD), and the d-spacing of the crystal was calculated based on 2θ. As shown in Figure 4b and Appendix A, four peaks at ~4.50 Å, ~4.30 Å, ~3.8 Å, and ~3.7 Å existed in GMS and GMC oleogels and the intensity varied with the MAG concentration. In the wide-angle region, the peak near 4.50 Å is generally considered to be the characteristic peak of the β-polymorph, 4.15 Å is the characteristic peak of the α-polymorph, and the peaks at 3.8 and 4.3 Å are the characteristic peaks of the β′-polymorph [12]. In other words, GMS and GMC oleogels featured both β and β′ polymorphic crystals. Unlike GMS and GMC oleogels, three peaks at ~4.50 Å, ~4.15 Å and ~3.91 Å were found in GML oleogels, indicating α polymorph existed in GML oleogels, which further proved that GML in soybean oil produced weaker gel network. This also validated our previous assumption that GML has different polymorphs. In addition, the intensity of each peak increased with the increasing MAG concentration. Yang et al. found that the increased gelator concentration contributes to self-sorting and rearrangement, resulting in better crystalline quality of the oleogels, thus showing higher strength on the XRD pattern [14]. This also explained why the oleogels made with higher MAG concentrations had higher hardness and moduli in the above results. 

### 3.6. Effects of Temperature on Crystallization Behavior

A polarized light microscope equipped with a thermostat was used to observe the morphology of crystals in oleogels under different temperatures. The number of crystals decreased with the increasing temperature, and when the temperature reached 80 °C, the crystals completely disappeared in the field of view (Figure 5). A gradual increase in the number of crystals was re-observed when the temperature decreased from 80 °C to 20 °C. At the same time, the visual appearance of oleogels during the heating and cooling processes was recorded, as shown in Appendix A. The conversion between solid and liquid oleogels could be observed during heating and cooling, which further confirmed the thermo-reversibility of MAG oleogels.

The effect of cooling rates on the microstructure of the oleogels is shown in Figure 5b. It can be observed that the size of the crystal aggregates decreased, and the number of particles increased with higher cooling rates, which might be attributed to higher cooling rates facilitating faster nucleation, resulting in smaller crystals [41,42]. Nevertheless, the basic morphology of the crystal aggregates in oleogels was independent of temperature cooling rates, which is consistent with the finding of Wettlaufer et al. [41].

## 4. Conclusions

The work focused on the gelling capacity of different fatty acid glycerides and the influence factors on the crystallization behavior of fatty acid glycerides-based oleogels. It is difficult for GMO and GDO to form solid-like oleogels. The results of PLM, DSC, and XRD showed that GMS and GMC oleogels exhibited smaller crystal size with uniform distribution, higher crystallinity, and the presence of β and β′ crystals, which contributed to the formation of more stable oleogels with higher OBC, hardness, thermal stability, and higher resistance to deformations. GML oleogels have larger clusters and α crystals, making their structure softer. The addition of more MAG to the MAG-oil systems produced a denser get network, resulting in higher OBC, hardness, and viscoelastic properties of oleogels. Interestingly, GMC-based oleogels presented similar properties as GMS-based oleogels. Temperature is also an important factor affecting the crystallization behavior of MAGs in oleogels. Overall, GMS and GMC are better choices than GML, and compared with unsaturated glycerides, saturated glycerides are preferred in the preparation of oleogels. Hence, the combination of MAGs parameters (i.e., chain length, molecular weight, type, and concentration of MAGs) and temperature control can be a promising way to tailor the crystallization behavior and physical properties of oleogels.

## Figures and Tables

**Figure 1 foods-12-00345-f001:**
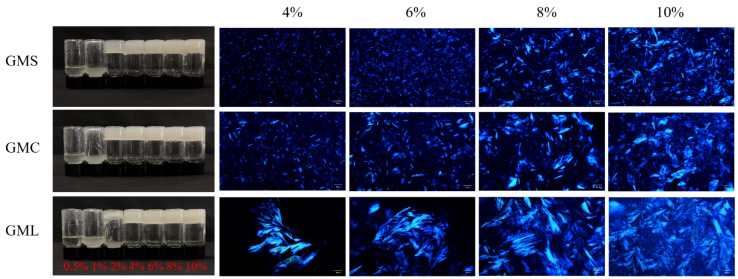
Visual appearance and polarized light microscopy images of different monoglyceride oleogels with different concentrations. The scale bar is 50 μm.

**Figure 2 foods-12-00345-f002:**
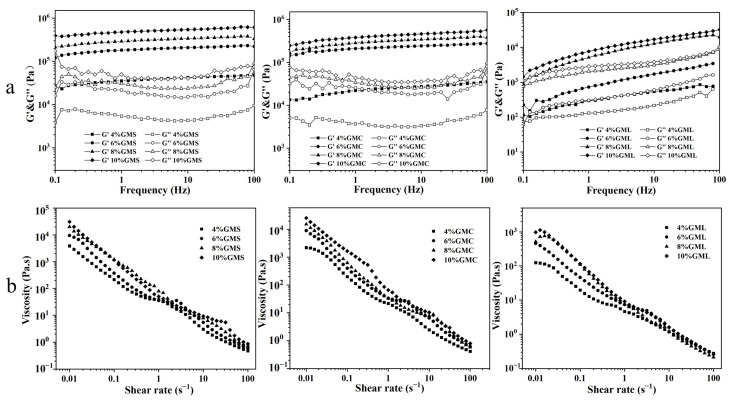
The behavior of the storage modulus (G′) and loss modulus (G″) in frequency sweep (**a**) and the shear-rate dependence of the viscosity of the oleogel samples (**b**).

**Figure 3 foods-12-00345-f003:**
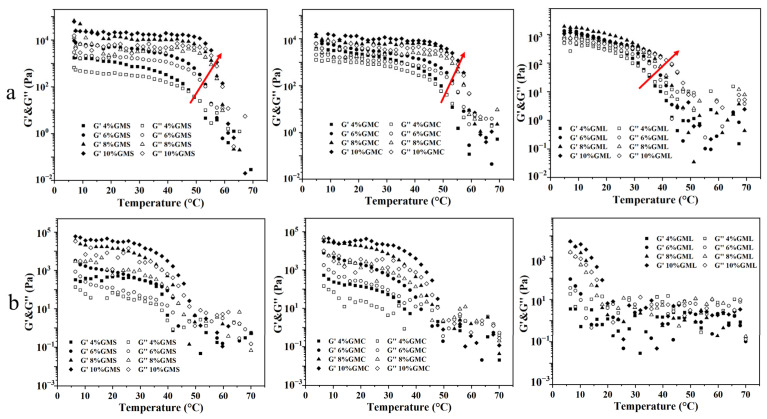
The storage modulus G′ and loss modulus G″ values during heating (**a**) and cooling (**b**) process. The red arrow indicates the direction in which the intersection of G′ and G″ moves from low to high MAG concentrations.

**Figure 4 foods-12-00345-f004:**
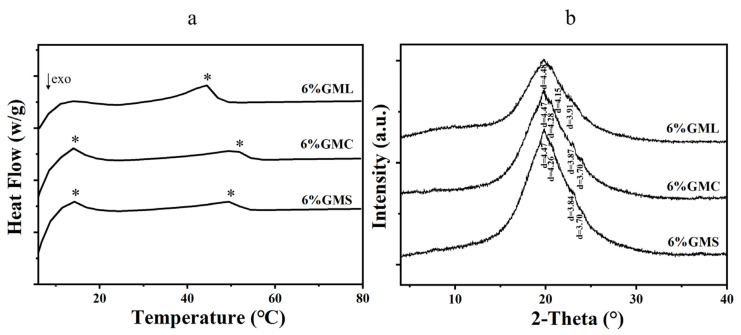
DSC Patterns (**a**) and X-ray Diffraction Patterns (**b**) of oleogels with 6% concentration. The * represents the endothermic peak of the oleogel samples.

**Figure 5 foods-12-00345-f005:**
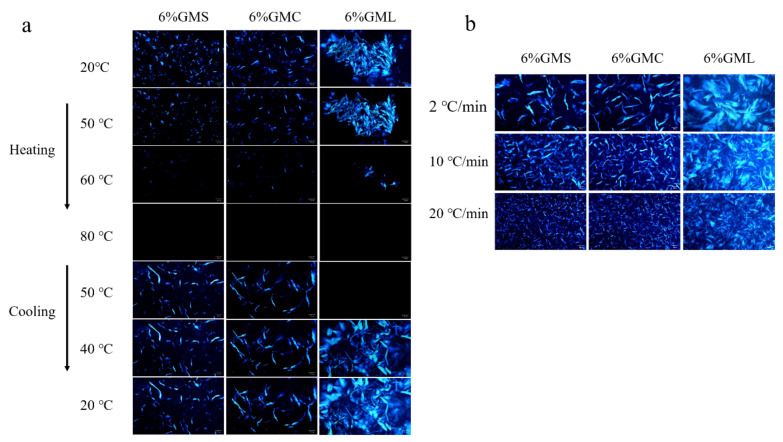
Polarized light microscopy images of oleogels during heating and cooling process with the rate of 2 °C/min (**a**) and with different cooling rates (2, 10 and 20 °C/min) (**b**). The scale bar is 50 μm.

**Table 1 foods-12-00345-t001:** The oil binding capacity (OBC) and hardness of the oleogels. Different lower-case letters (a, b, c, d) mean significant differences (*p* < 0.05) between different concentrations (the same MAG).

Sample	OBC (%)			Hardness (g)		
	GMS	GMC	GML	GMS	GMC	GML
4%	79.88 ± 1.18 ^a^	77.06 ± 0.74 ^a^	46.51 ± 2.13 ^a^	20.29 ± 1.86 ^a^	16.15 ± 0.51 ^a^	7.73 ± 0.45 ^a^
6%	93.61 ± 0.55 ^b^	85.11 ± 0.25 ^b^	58.39 ± 1.88 ^b^	47.47 ± 4.83 ^b^	48.62 ± 0.69 ^b^	18.98 ± 0.44 ^b^
8%	98.22 ± 0.50 ^c^	92.10 ± 1.95 ^c^	65.07 ± 0.86 ^c^	92.16 ± 4.19 ^c^	77.59 ± 3.18 ^c^	32.24 ± 2.19 ^c^
10%	98.06 ± 0.21 ^c^	93.75 ± 2.13 ^c^	74.85 ± 2.35 ^d^	106.25 ± 2.45 ^d^	98.75 ± 1.21 ^d^	40.23 ± 0.56 ^d^

## Data Availability

Data is contained within the article or Appendix A.

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
