# Peer review of "Crystallization Behavior and Physical Properties of Monoglycerides-Based Oleogels as Function of Oleogelator Concentration"

_foods, 2023, doi:10.3390/foods12020345_

Round 1

Reviewer 1 Report

Ms. Ref. No.:  foods-2099057

Title: Crystallization behavior and physical properties of mono-2 glycerides-based oleogels

 Comments and Suggestions from the Reviewer for improvement of the paper:

This manuscript proposes the development of oleogels as promising replacers for hydrogenated vegetable oils. The crystallization behavior and physical properties of oleogels were investigated by using different analysis techniques as a function of the chain length and concentration of fatty acid glycerides.

Overall, the paper will clearly benefit from an extended English revision. Along the manuscript is possible to find sentences without action verb and some misspelled words that may cause some confusion to the reader.

Any corelative conclusions were presented from the data achieved from the different analysis techniques. The establishment of those relations may allow to achieve significative conclusions or to design possible hypothesis.  

5/42 of the cited references have more than 10 years. Consider replacing it for more recent ones.

Particularly, some questions should be addressed along the manuscript, to increase the relevance of the publication.

Considering the crystals formed on the proposed oleogel systems, are the morphologies of the structures similar to the ones earlier reported in the literature?

The authors make reference to other works reporting differences in crystallization behaviors and the features that may influence it, still no comparison was made between the new reported crystal morphologies and the earlier reported ones. As well, a comparison between the fundamental morphological properties of the achieved crystals is required and its relation with the studied influence parameters (e.g. fatty acid glycerides chain lengh, concentration…). 

On the DSC spectrum, GML only present one peak in opposition to the two peaks phenomenon, observed for GMS and GML. The authors should present a hypothesis to justify this observation. As well an explanation about how this phenomenon may influence the values achieved for enthalpy should be presented.

What is the influence of the increase of crystallinity in the oleogel performance?

What is the relation between the increased crystallinity of the GML observed by DSC analysis and the XRD results?

How can the results be correlated?

The authors should clarify what is the influence of the increase of crystallinity on the features of the developed oleogel systems.

What is the influence of the heating and colling rates on the mechanical/rheological properties of the materials?

What is the advantage of developing a thermo-reversible system?

What is the specific application envisioned for the proposed oleogels?

Which is the preferential composition envisioning it? Why?

A better discussion should be included about the results achieved from the different characterization techniques employed, and establish a relation between the properties presented by the developed materials and the studied variables. A clear and supported conclusion should be included on the conclusions section.

The author does not make any reference along the manuscript, not even in the conclusion section, about the advantages of one of the composition over the others. There is no indication, about which is the best sample/composition or the required features for the envisioned material application.

In summary, this study shows some interesting aspects. Hence, I suggest MAJOR REVISION to the manuscript, still I recommend consideration after serious improvements.

Author Response

Manuscript Number: foods-2099057

Title: Crystallization behavior and physical properties of monoglycerides-based oleogels

Authors: Yingzhu Zhang a, Jinqi Xu a, Cuie Tang a,b,* and Yan Lia,b,c,*

Foods

Responses to Reviewer’s comments:

  • Again, we appreciate the reviewers for their worthy comments and suggestions on our manuscript. We have made the changes suggested by the reviewers’ and have included a detailed list of responses below. All the revised parts are highlighted in red. We hope our manuscript will now be suitable for publication. Your consideration of this revised version is appreciated.

Reviewers' comments:

Comments of editor and reviewers:

Response to Reviewer 1 Comments

Comments and Suggestions from the Reviewer for improvement of the paper:

This manuscript proposes the development of oleogels as promising replacers for hydrogenated vegetable oils. The crystallization behavior and physical properties of oleogels were investigated by using different analysis techniques as a function of the chain length and concentration of fatty acid glycerides.

Overall, the paper will clearly benefit from an extended English revision. Along the manuscript is possible to find sentences without action verb and some misspelled words that may cause some confusion to the reader.

Any corelative conclusions were presented from the data achieved from the different analysis techniques. The establishment of those relations may allow to achieve significative conclusions or to design possible hypothesis. 

5/42 of the cited references have more than 10 years. Consider replacing it for more recent ones.

Overall response: We have checked the language and spelling mistakes. Some of the refereces are the original findings, which we have to cite them.

Particularly, some questions should be addressed along the manuscript, to increase the relevance of the publication.

Point 1: Considering the crystals formed on the proposed oleogel systems, are the morphologies of the structures similar to the ones earlier reported in the literature?

Response 1: Eexcept GMC, the other MGs have been well reported in the literature, so the obscerved crystal morphologies of oleogels are similar to the reported results in previous studies. Pan et al. reported that the crystal morphologies of the oleogels formed by GML were large fiber-like, and Valoppi et al. reported the needle-like GMS oleogel crystals, which were consistent with the results of this study. However, the studies on GMC oleogels are relatively lacking. Based on our results, we found that GMC could also a potential oleo-gelator to form oleogels. GMC has the advantage of medium chain fatty acid.

Reference:

Pan, J.; Tang, L.; Dong, Q.; Li, Y.; Zhang, H. Effect of oleogelation on physical properties and oxidative stability of camellia oil-based oleogels and oleogel emulsions. Food Res Int. 2021, 140, 110057.

Valoppi, F.; Calligaris, S.; Barba, L.; Šegatin, N.; Poklar Ulrih, N.; Nicoli, M. C. Influence of oil type on formation, structure, thermal, and physical properties of monoglyceride-based organogel. Eur J Lipid Sci Tech. 2016, 119.

Point 2: The authors make reference to other works reporting differences in crystallization behaviors and the features that may influence it, still no comparison was made between the new reported crystal morphologies and the earlier reported ones. As well, a comparison between the fundamental morphological properties of the achieved crystals is required and its relation with the studied influence parameters (e.g. fatty acid glycerides chain lengh, concentration…).

Response 2: We have revised and supplemented in this paper, the details are in Line 171-173 and 177-182. Actually, we did not aim to report any new crystal morphologies. Only with higher MG concentration, more crystals were observed, which might cause the aggregations or large clusters, resulting dense gel network. Ln 185-186, we highlighted the finding of GMC crystals.

Point 3: On the DSC spectrum, GML only present one peak in opposition to the two peaks phenomenon, observed for GMS and GML. The authors should present a hypothesis to justify this observation. As well an explanation about how this phenomenon may influence the values achieved for enthalpy should be presented.

Response 3: It was reported that the melting and crystallization peak temperature of MGs was dependent on acyl chain length. That is the value of these peaks decreasing by the decreasing the chain length. So we also found that the peak value of GML is lower than that of GMS. Interestingly, GMC showed similar DSC curve with GMS, which indicated that GMC might be a proimsing oleogelator. We have revised the discussion in Line 277-296.

Point 4: What is the influence of the increase of crystallinity in the oleogel performance?

Response 4: The formation of MG-based oleogels is mainly due to the formation of MG crystals in the systems. Hence, the crystallinity of oleogel systems could reflect the self-assembly of MG moleculues in the oil systems, which causing the compact of three-dimensional network. Hence, we measured the crystallization bahevior of the oleogels.

Point 5: What is the relation between the increased crystallinity of the GML observed by DSC analysis and the XRD results?

Response 5: DSC measurement indicated the thermal bahavior of crystals, including crystallization and melting profiles of oleogels. XRD measurement indicated the polymorphism of crystals in the oleogels. The polymorphism of crystals would affect the thermal behavior of crystals.

Point 6: How can the results be correlated?

Response 6: DSC and XRD results indicated the poorer stability and weaker network formed by GML inside the oleogels, which was supported by the microstructure, OBC and hardness results. Overall, GMS and GMC oleogels possess higher crystallinity, which caused higher heat resistance and better me-chanical properties than GML-based oleogels.

Point 7: The authors should clarify what is the influence of the increase of crystallinity on the features of the developed oleogel systems.

Response 7: DSC and XRD results indicated the poorer stability and weaker network formed by GML inside the oleogels, which was supported by the microstructure, OBC and hardness results. Overall, GMS and GMC oleogels possess higher crystallinity, which caused higher heat resistance and better me-chanical properties than GML-based oleogels. We have emphasized this information in the paper.

Point 8: What is the influence of the heating and colling rates on the mechanical/rheological properties of the materials?

Response 8: It is a good point. We found that oleogels had different thermal stability and rheological properties as a function of temperature. Hence, we tried to in situ characterize the change of morphology of crystals at different heating and cooling rates. But we failed to study the the influence of the heating and cooling rates on the mechanical/rheological properties of the materials. Since, we could not control the heating and cooling rate for the sample preparation. So we did not do it, which would our plan to carry out.

Point 9: What is the advantage of developing a thermo-reversible system?

Response 9: As a potential replacement of plastic oil, the solidity of oleogel at low temperature helps to easily shape or cream and the conversion to flow ability of oleogel at high temperature could provide the nature of liquid oil.

Point 10: What is the specific application envisioned for the proposed oleogels?

Response 10: Oleogel can be used to replace solid fat in various foods and to stabilize emulsions, it can also be used as delivery vehicles. The specific application of oleogels can be referred to the review literature of Zhao et al. In our subsequent studies, MAG based oleogels were used as solid lipids to co-stabilize water-in-oil emulsions with PGPR and finally to prepare low-fat margarine (the following figure).

Reference:

Zhao, W.; Wei, Z.; Xue, C. Recent advances on food-grade oleogels: Fabrication, application and research trends. Crit Rev Food Sci Nutr. 2021, 1-18.

Point 11: Which is the preferential composition envisioning it? Why?

Response 11: It it speculated that the combination of GMS and GMC is preferential selection. For single-component oleogels, GMS- and GMC-based oleogels had higher heat resistance and better mechanical properties than GML-based ones at the same monoglyceride content level, and in the study of Li et al., the combination of two monoglycerides with stronger gel strength showed better mechanical strength and stability, therefore we speculate that the combination of GMS and GMC is optimal choice.

Reference:

Li, J.; Guo, R.; Bi, Y.; Zhang, H.; Xu, X. Comprehensive evaluation of saturated monoglycerides for the forming of oleogels. Lwt. 2021, 151.

Point 12: A better discussion should be included about the results achieved from the different characterization techniques employed, and establish a relation between the properties presented by the developed materials and the studied variables. A clear and supported conclusion should be included on the conclusions section.

Response 12: We have revised and modified the discussion section.

Point 13: The author does not make any reference along the manuscript, not even in the conclusion section, about the advantages of one of the composition over the others. There is no indication, about which is the best sample/composition or the required features for the envisioned material application.

Response 13: According to the comment, GMS and GMC oleogels exhibit smaller crystal size, uniform distribution, higher crystallinity and the presence ofβandβ′crystals, which contributed to the formation of more stable oleogels with higher OBC, hardness, thermal stability and higher re-sistance to deformations. So GMS and GMC are better choices than GML. In addition, compared with unsaturated glycerides, saturated glycerides are the preferred choice. Finally, the appropriate concentration can be selected according to the requirements of the oleogel performance. We have highlighted this information in the revised paper.

Reviewer 2 Report

The work deals with the characterization of monoglyceride-based oleogels. It was writing in a clear and logical way. Only one thing, the novelty of the study should be stated more clear compared to previous works e.g., “Li, J., Guo, R., Bi, Y., Zhang, H., & Xu, X. (2021). Comprehensive evaluation of saturated monoglycerides for the forming of oleogels. LWT151, 112061.”

Author Response

Manuscript Number: foods-2099057

Title: Crystallization behavior and physical properties of monoglycerides-based oleogels

Authors: Yingzhu Zhang a, Jinqi Xu a, Cuie Tang a,b,* and Yan Lia,b,c,*

Foods

Responses to Reviewer’s comments:

  • Again, we appreciate the reviewers for their worthy comments and suggestions on our manuscript. We have made the changes suggested by the reviewers’ and have included a detailed list of responses below. All the revised parts are highlighted in red. We hope our manuscript will now be suitable for publication. Your consideration of this revised version is appreciated.

Reviewers' comments:

Response: We have read this paper during our work. This previous work mainly studied the effect of pure monoglyceride with three acyl chain length and mixture of monoglycerides on the properties and stability of oleogels. The molecular chain length and composition of monoglycerides were emphasized. While in the present work, we also selected different glycerides, but rather than only acyl chain length. The saturation of fatty acids is also considered. More importantly, we selected GMC with medium chain fatty acid. In the previous work, MG with long acyl chain length is usually used. The second difference is the concentration-dependance study. We studied the concentration of MG on the formation of oleogel and crystallization behavior.

The third difference is the in situ characterization of temperature (heating and cooling rate) effect on the formation of crystals in the oleogels.

Therefore, our work could provide more interesting findings. These findings are included in the revised abstract.

Reviewer 3 Report

In this paper the authors report on systems they name oleogels. I have several comments about the relevancy of the terminologies used.

1) In view of the OM pictures, these systems are not gels as is usually accepted. Gels are primarily networks of entities such as fibrils.

In language dictionaries a network is:” a large system of lines, tubes, wires, etc. that cross one another or are connected with one another”.

The authors can peruse the following paper about the discussion of this definition: Guenet, J.M.Physical Aspects of Organogelation: A Point of View Gels 2021, 7, 65 and in Guenet, J.M. Organogels: Thermodynamics, Structure, Solvent Role and Properties, Springer, N.Y. USA, 2016

While I can understand that the term gels as appears to be usual in the food industry for these systems, they must discuss this point in order to avoid any confusion, particularly on the gelation mechanism.

The fact that they observe G’> G’’ in rheological experiments is by no means a proof that these systems behave like gels. Indeed, this is also discussed in the above paper and in Gilsenan, PM; Ross-Murphy, SB Journal of Rheology 2000, 44, 871. The fact that G’’ increases at low frequencies while G’ decreases may suggest that G’’> G’ at much lower frequencies. Only relaxation experiments could discriminate between a gel and a paste.

See for instance Daniel et al. On the Definition of Thermoreversible Gels: Case of Syndiotactic Polystyrene. Polymer 35, 4243, 1994

So, the authors have to mention that these systems are conveniently designated as oleogels although they are not strictly speaking gels in view of the accepted definition.

2) About the critical gelation temperature. As discussed also by Guenet (see above), this cannot be considered a critical parameter as it depends on the temperature at which the system is cooled, and the way the system is prepared (slow cooling or rapid cooling). For instance, this concentration is likely to be higher at 30°C or by a rapid quench. The authors should rather use “onset gelation concentration”.

3) The DSC pattern at low temperature is strange. The so-called peak is observed while the DSC set-up is still in the process of equilibrating. Clearly, the authors must do runs at different heating rates, particularly at much lower rates. Also, the events occurring during the cooling process must be reported.

4) XRD patterns. First, the intensity should be plotted as a function of q (4π/λ sinθ/2) not θ. This allows one to compare with data obtained at different wavelengths as is the case when using synchrotron radiation. Second, the authors should use a software for deconvoluting the peaks. This would help them to assign peak positions. For instance, they consider a peak at d= 3.87 for 6% GMC but ignore a possible peak nearby at θ≈ 24°.

Author Response

Manuscript Number: foods-2099057

Title: Crystallization behavior and physical properties of monoglycerides-based oleogels

Authors: Yingzhu Zhang a, Jinqi Xu a, Cuie Tang a,b,* and Yan Lia,b,c,*

Foods

Responses to Reviewer’s comments:

  • Again, we appreciate the reviewers for their worthy comments and suggestions on our manuscript. We have made the changes suggested by the reviewers’ and have included a detailed list of responses below. All the revised parts are highlighted in red. We hope our manuscript will now be suitable for publication. Your consideration of this revised version is appreciated.

Reviewers' comments:

Comments of editor and reviewers:

Response to Reviewer 3 Comments

Point 1: In view of the OM pictures, these systems are not gels as is usually accepted. Gels are primarily networks of entities such as fibrils.

In language dictionaries a network is:” a large system of lines, tubes, wires, etc. that cross one another or are connected with one another”.

The authors can peruse the following paper about the discussion of this definition: Guenet, J.M.Physical Aspects of Organogelation: A Point of View Gels 2021, 7, 65 and in Guenet, J.M. Organogels: Thermodynamics, Structure, Solvent Role and Properties, Springer, N.Y. USA, 2016

While I can understand that the term gels as appears to be usual in the food industry for these systems, they must discuss this point in order to avoid any confusion, particularly on the gelation mechanism.

The fact that they observe G’> G’’ in rheological experiments is by no means a proof that these systems behave like gels. Indeed, this is also discussed in the above paper and in Gilsenan, PM; Ross-Murphy, SB Journal of Rheology 2000, 44, 871. The fact that G’’ increases at low frequencies while G’ decreases may suggest that G’’> G’ at much lower frequencies. Only relaxation experiments could discriminate between a gel and a paste.

See for instance Daniel et al. On the Definition of Thermoreversible Gels: Case of Syndiotactic Polystyrene. Polymer 35, 4243, 1994

So, the authors have to mention that these systems are conveniently designated as oleogels although they are not strictly speaking gels in view of the accepted definition.

Response 1: We do agree with this comment. In the oleogel systems, there are also some definitions and literatures. Giacomozzi et al. say “Oleogels are structured oil systems that can be defined as semisolid materials obtained by entrapping liquid oil into a three-dimensional network”. Zhao et al. writtrn in their article that “Oleogel, which is composed of lipophilic liquid (generally vegetable oil) and a small amount of oleogelators, is a sort of thermoreversible semi-solid lipid mixture with robust viscoelastic properties”. At present, the definition of oleogels is relatively uniform, and the research about oleogels is also very extensive and mature, such as “Comprehensive evaluation of saturated monoglycerides for the forming of oleogels” and “Role of the oil on glyceryl monostearate based oleogels”. Therefore, consistent with previous studies, we believe that the oleogels are a gel system stabilized by a three-dimensional network structure formed by the gelator. It is written in our article that “the storage moduli (G′) were higher than the loss moduli (G″) without overlapping in the frequency range, indicating that all oleogels were viscoelastic materials with solid-like behavior” is consistent with the study of Aliasl et al., Li et al. and Thakur et al.

Reference:

Giacomozzi, A. S.; Palla, C. A.; Carrin, M. E.; Martini, S. Physical Properties of Monoglycerides Oleogels Modified by Concentration, Cooling Rate, and High-Intensity Ultrasound. J Food Sci. 2019, 84, 2549-2561.

Zhao, W.; Wei, Z.; Xue, C. Recent advances on food-grade oleogels: Fabrication, application and research trends. Crit Rev Food Sci Nutr. 2021, 1-18.

Aliasl Khiabani, A.; Tabibiazar, M.; Roufegarinejad, L.; Hamishehkar, H.; Alizadeh, A. Preparation and characterization of carnauba wax/adipic acid oleogel: A new reinforced oleogel for application in cake and beef burger. Food Chem. 2020, 333, 127446.

Li, J.; Guo, R.; Bi, Y.; Zhang, H.; Xu, X. Comprehensive evaluation of saturated monoglycerides for the forming of oleogels. Lwt. 2021, 151.

Thakur, D.; Singh, A.; Prabhakar, P. K.; Meghwal, M.; Upadhyay, A. Optimization and characterization of soybean oil-carnauba wax oleogel. Lwt-Food Science and Technology. 2022, 157.

Point 2: About the critical gelation concentration. As discussed also by Guenet (see above), this cannot be considered a critical parameter as it depends on the temperature at which the system is cooled, and the way the system is prepared (slow cooling or rapid cooling). For instance, this concentration is likely to be higher at 30°C or by a rapid quench. The authors should rather use “onset gelation concentration”.

Response 2: According to reviewer’s comment, we replaced this description of critical gelation concentration by ‘onset gelation concentration’.  In some references, they also used the critical gelation concentration by inverted tube method, which we referred to. For example, Zheng et al described in their study ”The concentration of GML was varied from 0% (w/w) to 40% (w/w) to find the critical gelation concentration (CGC). The phenomenon of gelation was confirmed by the inverted tube method”.

Zheng, H.; Deng, L.; Que, F.; Feng, F.; Zhang, H. Physical characterization and antimicrobial evaluation of glycerol monolaurate organogels. Colloid Surface. 2016, 502, 19-25.

Behera, B.; Sagiri, S. S.; Pal, K.; Srivastava, A. Modulating the physical properties of sunflower oil and sorbitan monopalmitate-based organogels. Journal of Applied Polymer Science. 2013, 127, 4910-4917.

Point 3: The DSC pattern at low temperature is strange. The so-called peak is observed while the DSC set-up is still in the process of equilibrating. Clearly, the authors must do runs at different heating rates, particularly at much lower rates. Also, the events occurring during the cooling process must be reported.

Response 3: The DSC parameters refer to the reasearch of Yang et al. and Zhao et al. To do the DSC pattern, we first ran the samply at higher heating rate (30 °C/min) and then used the rate of 10 °C/min. The whole DSC pattern is shown as follow. In the present work, we focued on the 2nd heating run.

Reference:

Yang, S.; Li, G.; Saleh, A. S. M.; Yang, H.; Wang, N.; Wang, P.; Yue, X.; Xiao, Z. Functional Characteristics of Oleogel Prepared from Sunflower Oil with β-Sitosterol and Stearic Acid. JACS. 2017, 94, 1153-1164.

Zhao, M.; Lan, Y.; Cui, L.; Monono, E.; Rao, J.; Chen, B. Formation, characterization, and potential food application of rice bran wax oleogels: Expeller-pressed corn germ oil versus refined corn oil. Food Chem. 2020, 309, 125704.

Point 4: XRD patterns. First, the intensity should be plotted as a function of q (4π/λ sinθ/2) not θ. This allows one to compare with data obtained at different wavelengths as is the case when using synchrotron radiation. Second, the authors should use a software for deconvoluting the peaks. This would help them to assign peak positions. For instance, they consider a peak at d= 3.87 for 6% GMC but ignore a possible peak nearby at θ ≈ 24°.

Response 4: We have revised in this paper, and the details are in Figure 4 and section 3.5. But for XRD analysis of oleogels, the intensity is plotted as a function of 2θ, which was refered from the following literatures.

Li, J.; Guo, R.; Bi, Y.; Zhang, H.; Xu, X. Comprehensive evaluation of saturated monoglycerides for the forming of oleogels. Lwt. 2021, 151.

Ferro, A. C.; Okuro, P. K.; Badan, A. P.; Cunha, R. L. Role of the oil on glyceryl monostearate based oleogels. Food Res Int. 2019, 120, 610-619.

Guo, S.; Song, M.; Gao, X.; Dong, L.; Hou, T.; Lin, X.; Tan, W.; Cao, Y.; Rogers, M.; Lan, Y. Assembly pattern of multicomponent supramolecular oleogel composed of ceramide and lecithin in sunflower oil: self-assembly or self-sorting? Food Funct. 2020, 11, 7651-7660.

Round 2

Reviewer 3 Report

We all agree that a gel is a network. So the defintion of a network, as I mentioned previously, provided in the dictionary is

a large system of lines, tubes, wires, etc. that cross one another or are connected with one another

Clearly, from the OM pictures these systems have not at all the aspect of a network! They rather consist of intertwined spherulites.

Also, as I explained the rheological experiments cannot be conclusive, especially as one can see an upturn of G'' in the low frequency range. Again, only a relaxation experiment could definitely allow one to conclude whether one is dealing with an elastic system or not. In view of my personal experience, these systems most probably behave like a paste.

I am not asking to change the title but simply to discuss that point in order not to throw any confusion in the domain.

Author Response

Manuscript Number: foods-2099057

Title: Crystallization behavior and physical properties of monoglycerides-based oleogels as function of oleogelator concentration

Authors: Yingzhu Zhang a, Jinqi Xu a, Cuie Tang a,b,* and Yan Lia,b,c,*

Foods

Responses to Reviewer’s comments:

  • Again, we appreciate the reviewers for their worthy comments and suggestions on our manuscript. We have made the changes suggested by the reviewers’ and have included a detailed list of responses below. All the revised parts are highlighted in red. We hope our manuscript will now be suitable for publication. Your consideration of this revised version is appreciated.

Reviewers' comments:

Response:

Thanks so much for your clear explanation. Now we can well understand what your comment means. We have carefully read the references and do agree the point that the definition of gels should be clear. Usually, we used the test tube to test the formation of a gel. But except the phase transition, gel system should meet more criteria, proposed in the recommended references. According to lots of similar references, the paper called this kind of spherulites structure as the gel structure. Also the value of elastic modulus is higher than that of viscous modulus, so we called the system as the gel system. In our work, OM pictures showed that only GML sample was more like the spherulitic structure at low GML concentration, which could be like a paste. For GMS and GMC, the structure is more like a network structure according the recommend reference. At high concentration, the crystalls of MGD would connect and intertwine together which causes the increase of gel hardness. We highlighted this information in the revised paper.

For the rheological tests, we did not do the relaxation experiments in the present work. But it will be a useful mechanical test to definite the gel systems. In the present work, the upturn of of G'' in the low frequency range is more obvious in the GML systems. It might be due to the weak interaction of GML crystals. We highlighted this observation in the analysis of rheological tests.
